# Cardiac Magnetic Resonance Imaging as a Risk Stratification Tool in COVID-19 Myocarditis

**DOI:** 10.3390/diagnostics14080790

**Published:** 2024-04-10

**Authors:** Olga Nedeljkovic-Arsenovic, Arsen Ristić, Nemanja Đorđević, Milenko Tomić, Gordana Krljanac, Ruzica Maksimović

**Affiliations:** 1Department of Magnetic Resonance Imaging, Centre for Radiology, University Clinical Centre of Serbia, Pasterova 2, 11000 Belgrade, Serbia; dr.ruzica.maksimovic@gmail.com; 2Faculty of Medicine, University of Belgrade, Dr Subotica 8, 11000 Belgrade, Serbia; gkrljanac@gmail.com; 3Clinic for Cardiology, University Clinical Centre of Serbia, 11000 Belgrade, Serbia; arsen.ristic@med.bg.ac.rs (A.R.); nzdjordj@gmail.com (N.Đ.); tomic_milenko@outlook.com (M.T.)

**Keywords:** myocarditis, COVID-19 infection, cardiac magnetic resonance, late gadolinium enhancement and prognosis

## Abstract

The aim of this retrospective study was to identify myocardial injury after COVID-19 inflammation and explore whether myocardial damage could be a possible cause of the persistent symptoms following COVID-19 infection in previously healthy individuals. This study included 139 patients who were enrolled between January and June 2021, with a mean age of 46.7 ± 15.2 years, of whom 68 were men and 71 were women without known cardiac or pulmonary diseases. All patients underwent clinical work-up, laboratory analysis, cardiac ultrasound, and CMR on a 1.5 T scanner using a recommended protocol for morphological and functional assessment before and after contrast media application with multi-parametric sequences. In 39% of patients, late gadolinium enhancement (LGE) was found as a sign of myocarditis. Fibrinogen was statistically significantly higher in patients with LGE than in those without LGE (4.3 ± 0.23 vs. 3.2 ± 0.14 g/L, *p* < 0.05, respectively), as well as D-dimer (1.8 ± 0.3 vs. 0.8 ± 0.1 mg/L FEU). Also, troponin was statistically significantly higher in patients with myocardial LGE (13.1 ± 0.4 ng/L) compared to those with normal myocardium (4.9 ± 0.3 ng/L, *p* < 0.001). We demonstrated chest pain, fatigue, and elevated troponin to be independent predictors for LGE. Septal LGE was shown to be a predictor for arrhythmias. The use of CMR is a potential risk stratification tool in evaluating outcomes following COVID-19 myocarditis.

## 1. Introduction

Myocarditis is an inflammatory process affecting the myocardium that may be caused by infective and toxic agents or immune-mediated factors. Myocarditis is the third most common cause of sudden cardiac death and has been associated with 5–12% of sudden cardiac deaths in young athletes [1]. Diagnosis by clinical assessment is not always reliable because myocarditis has a wide spectrum of clinical manifestations. Endomyocardial biopsy is the gold standard for the diagnosis of myocarditis, but it carries its own risks as an invasive procedure [2]. Recent research suggests that COVID-19 infection leads to endothelial damage of multiple organs, including the kidneys, heart, brain, and blood vessels, followed by systemic inflammation [3]. Besides affecting the respiratory system, COVID-19 infection has adverse effects on the cardiovascular system and has been associated with myocardial injury, causing ischemia, inflammation, or myocarditis [4]. The development of myocarditis associated with COVID-19 infection leads to considerable morbidity and mortality, along with a degree of myocardial injury that can be assessed using cardiac magnetic resonance imaging (CMR). Cardiac injury as a result of COVID-19 infection has been associated with worse prognosis [3].

CMR imaging allows comprehensive assessment of myocardial function and tissue characterization in patients with cardiovascular disease. Information provided by overall CMR analysis, including left ventricular ejection fraction, can be a useful tool to assess prognosis in cardiac disease in addition to conventional risk factors [5]. Therefore, CMR parameters of myocardial tissue damage and cardiac function are increasingly recognized in order to improve treatment and therapy in this population.

CMR as a non-invasive imaging modality has been introduced, and its use has been established for detecting myocarditis. CMR allows a non-invasive diagnosis of the disease but does not allow the underlying cause to be distinguished [6]. It has a high positive predictive value, making a diagnosis in up to 79% of pathologically proven myocarditis cases [7].

Nowadays, CMR findings of myocarditis in patients post-COVID-19 infection have become a subject of interest. The majority of patients who have cardiovascular complications related to COVID-19 infection have myocarditis as the primary cause of cardiac dysfunction [8]. COVID-19 infection is associated with myocardial injury in a significant number of patients. Myocardial injury in hospitalized patients with COVID-19 is associated with a worse prognosis [9]. According to the literature data, many patients with COVID-19 infection had myocardial injury. A retrospective single-center study carried out in Wuhan demonstrated a significant correlation between myocardial injury and mortality and worse outcomes even when compared with patients with previous cardiovascular disease [10]. Myocardial involvement in COVID-19 disease is associated with a worse prognosis, but isolated myocarditis is not necessarily a marker of poor prognosis. Myocardial damage in the form of edema, LGE phenomenon/fibrosis, and pericardial inflammation after COVID-19 infection was seen using CMR.

CMR imaging has benefits in assessing myocarditis using Lake Louise criteria (LLC) established in 2009, which include the presence of edema, hyperemia, and necrosis/or fibrosis [11]. The original LLC were modified in 2018 since several limitations were progressively discovered. As per the 2018 LLC, the diagnosis of CMR-based myocarditis should include at least one T1-based criterion (increased myocardial T1 relaxation times, extracellular volume, or late gadolinium enhancement—LGE) plus at least one T2-based criterion (increased myocardial T2 relaxation times, visible regional high T2 signal intensity representing edema on T2 STIR, or increased T2 signal intensity ratio) [12,13] (Figure 1).

### Purpose

The purpose of this study was to identify myocardial injury in patients after COVID-19 infection and to explore the relationship between clinical and laboratory parameters and the degree of myocardial damage. Also, we intended to explore whether myocardial injury and inflammation could be a possible cause of the persistent symptoms following COVID-19 infection in previously healthy individuals.

## 2. Materials and Methods

This study is a retrospective, single-center analysis conducted at the University Clinical Centre of Serbia and was performed in concordance with the Helsinki Declaration and International Conference on Harmonization of Good Clinical Practice. Written informed consent of patients was obtained after providing them all with information regarding this study and the potential risks of participation.

### 2.1. Study Participants

This study included 139 patients who were enrolled between January and June 2021. The mean age of participants was 46.7 ± 15.2 years; 68 were men (49%) and 71 were women (51%). This study included patients who were previously healthy individuals without any treated cardiac or pulmonary disease. They had COVID-19 infection and persistent symptoms following COVID-19 infection. Those symptoms were predominantly cardiovascular symptoms such as persistent fatigue, exertional dyspnea, chest pain, and arrhythmias after COVID-19 convalescence. Persistent clinical symptoms were the main reason for CMR in order to evaluate the presence of myocardial damage or myocarditis as a possible explanation of the mentioned symptoms. All patients underwent clinical work-up, laboratory analysis, EKG, cardiac ultrasound, and CMR.

Patients with chronic COVID syndrome (CCS) had a negative polymerase chain reaction test result and had experienced resolution of acute COVID-19 symptoms for at least 2 weeks before CMR was performed.

### 2.2. CMR Protocol

CMR was performed using a standard protocol for morphological and functional assessment, LGE, as well as T1 and T2 mapping using a modified Look–Locker inversion recovery (MOLLI) sequence before and after contrast media application. Before CMR scanning, the weight and height of the patients were measured, and they were informed about the procedure. Examinations were performed using a 1.5 T scanner (AvantoFit, Siemens Magnetom, Erlangen, Germany). During the evaluation, patients were placed lying down in the supine position, headfirst, with ECG pads placed on the chest.

Electrocardiogram-gated steady-state free-precession (SSFP) cine images were acquired in short-axis, two-chamber, three-chamber, and four-chamber views for functional analysis. All images were obtained using retrospective gating during a gentle expiratory breath hold. Short-axis cine images were acquired as a stack from the mitral valve plane through the apex covering the entire ventricles; slice thickness was 8 mm, and field of view was 360 × 306 mm. T2-weighted with fat- suppression sequences in short-axis and 4-chamber views were performed for visualization of myocardial edema.

T1 mapping was performed with a modified MOLLI sequence, with a 3(3)3(3)5 sampling pattern, acquired before and 15 min following bolus contrast administration in 3 short-axis images (basal, mid-ventricular, and apical levels) and in a 4-chamber view with variable inversion preparation time, during the same cardiac phase at late diastole using the same imaging parameters. Acquisition parameters included echo time/repetition time = 1.13/279.84 ms, field of view = 360 × 306 mm, flip angle = 35°, matrix size = 256× 168, GRAPPA = 2, 36 reference lines, cardiac delay time TD =504 ms, interpolation = 0.7 × 0.7, inversion-time increment of 180 ms, and partial-phase Fourier 7/8. For myocardial T2 mapping, a six-echo gradient spin-echo sequence was applied.

LGE images were acquired after 8–10 min of contrast application using a phase-sensitive inversion recovery sequence (PSIR). Before patients were scanned, a time-to-inversion (TI) scout was performed in order to select the optimal inversion time for delayed enhancement. PSIR was performed in SAX view covering the whole ventricle with a slice thickness of 6 mm, and also in 2-, 3-, and 4-chamber views. LGE images were obtained at the different planes for the presence or the absence of enhancement in each segment. For contrast enhancement, a bolus of 0.2 mmol of Gadobutrol (Gadovist, Bayer Inc., Toronto, ON, Canada) per kilogram of body weight was administered.

### 2.3. Image Analysis

Image analysis was performed using the dedicated software Syngo via by Siemens and reviewed by two experienced radiologists, one with over 20 years of experience in cardiac MRI and the other with 3 years of experience. Using Argus (Siemens Healthcare, Erlangen, Germany), LV short-axis epicardial and endocardial borders were manually contoured at end diastole and end systole for determining EDV, ESV, SV, EF, and myocardial mass in short-axis images [14]. Parameters were recorded in absolute numbers and indexed by body surface area. Images were acquired in traditional multi-breath-hold cine CMR sequences.

The presence of focal areas of regional high signal intensities in a non-ischemic distribution pattern on T2-weighted sequence with fat suppression and on LGE images was visually assessed by two radiologists. Quantitative markers of myocardial edema were calculated by T2 mapping values. Reference cut-off values for T1 and T2 mappings were measured by ROI. The cut-offs for the diagnosis of myocardial inflammation were greater than or equal to 1060 ms for myocardial T1 relaxation times and greater than or equal to 55 ms for myocardial T2 relaxation times.

### 2.4. Statistical Analysis

SPSS Statistics version 27 (IBM) was used for all statistical analyses. Participant characteristics are given as means ± standard error or as absolute frequencies with percentages. Data were checked for normal distribution by using the Shapiro–Wilk test. Continuous variables were compared using Student’s *t*-test or the Mann–Whitney *U* test and categorical variables were compared using the Chi^2^ test. For comparison of continuous inter-individual variables, Student’s *t*-test or the Mann–Whitney test was used depending on the data distribution. For intra-individual comparisons, the Wilcoxon rank test was used.

For an exploratory analysis of potential predictors for cardiac fibrosis, laboratory parameters, clinical symptoms, and native T1 maps were first tested in a single-variable analysis and then, if significantly different, in multivariate analysis. Hazard ratios with 95% CIs were calculated with Cox models. All tests were two-sided, and *p* < 0.05 was considered indicative of a statistically significant difference.

## 3. Results

### 3.1. Patient Characteristics

A total of 139 participants were included in this retrospective study and statistical evaluation. There were no significant differences between men and women regarding age (*p* = 0.56).

The median time between initial COVID-19 diagnosis and CMR was 135 days (range 15–378 days) (minimum 15 days, max 378 days). Participants undergoing CMR exam following recovery of COVID-19 infection reported dyspnea (58 pts, 41.7%), fatigue (82 pts, 59%), chest pain (48 pts, 34.5%), and arrhythmias (54 pts, 39%) (Table 1).

### 3.2. CMR Findings

In this population, 85 pts (61.2%) had a normal myocardium on CMR, while in 54 patients (38.8%), LGE was found as a sign of myocarditis sequelae. LGE can persist forever as a sign of definite fibrosis. It depends on accompanying edema in the myocardium. If there is edema in the myocardium, there is a bigger chance that fibrosis will disappear [15]. As we already mentioned, our CMR was performed the earliest 15 days after the infection, and the mean time to complete CMR was 135 days, so the majority of patients had CMR performed in the subacute and chronic phases. We did not obtain statistical significance in any separate segment of the left ventricle since the number of patients in those subgroups was insufficient. Therefore, in order to increase the sample size and according to the clinical practice, we divided patients into groups by localization of LGE distribution in the left ventricle wall (septal wall, lateral wall, more than three segments, and other localizations). Segmentation of the left ventricle was performed manually in this study, although in the study by Zotti et al. [16], promising results were reported using neural networks, both regarding the accuracy and time of processing. Also, automatic segmentation of the left atrium (LA) in magnetic resonance images is of great significance for studying the structure of the left atrium in order to facilitate the diagnosis and treatment of atrial fibrillation [17].

Findings of myocarditis were more often seen in male than in female patients (61% vs. 39%) with a statistical significance regarding sex (*p* = 0.032, *p* < 0.05), while there were no significant differences based on age (*p* = 0.566). Statistically significant higher values of T1 mapping were seen in the LGE group (1103.9 ± 17.2 ms) than in patients with normal myocardium (1063.5 ± 9.7 ms) (t = −2.2, *p* = 0.033).

### 3.3. Laboratory Parameters

Fibrinogen levels were statistically significantly higher in patients with LGE than in those without LGE (4.3 ± 0.23; 3.2 ± 0.14 g/L, respectively; *p* < 0.001, t = −4). Also, D-dimer levels were higher in patients with LGE (1.8 ± 0.3 mg/L FEU) compared to those with no myocardial LGE (0.8 ± 0.1 mg/L FEU). This was statistically significant (*p* = 0.005, Z = −3.9). Troponin was another laboratory parameter that was statistically significantly higher in patients with myocardial LGE (13.1 ± 0.4 ng/L) compared to those with normal myocardium (4.9 ± 0.3 ng/L) with *p* < 0.001, Z = −9.5. Higher values of creatine kinase (CK) were found in the LGE group at 143.9 ± 12.4 u/L compared to the non-LGE group, which had a mean value of CK of 112.8 ± 6.9 u/L. There were statistically significant differences between those values (*p* < 0.05, *p* = 0.042, Z = −1.6).

### 3.4. Clinical Symptoms

Patients with fatigue as a symptom of COVID-19 infection had a 22-fold risk for LGE (Chi^2^ 41.2, *p* < 0.001), and 50 (61%) of them had a myocardial injury or LGE, while 32 (39%) had normal myocardium.

Regarding dyspnea, there were no significant differences between groups with LGE and without LGE (Chi^2^ 1.5, *p* = 0.213). In patients with dyspnea, LGE was seen in 19 pts (32.8%), while 39 pts (67.2%) had normal CMR findings.

Chest pain was demonstrated to be a predictor for LGE (Chi^2^ 9.3, *p* < 0.01, *p* = 0.002). Of patients who had symptoms of chest pain, 27 had LGE (56.3%), while 21 pts had no LGE (43.7%).

We demonstrated in multivariate analysis that from all those parameters that were statistically significant in univariate analysis, only chest pain, fatigue, and troponin elevation were independent predictors for LGE (Chi^2^ 112.6, *p* < 0.001, B = −10.6) adjusted for age and sex (Table 2).

### 3.5. LGE Localization

Regarding patients with LGE and myocardial damage (n = 54), a six-fold increased risk for arrhythmias was noted in comparison with those without myocardial fibrosis. Related to localization of LGE in patients with arrhythmias, only one patient had fibrosis in the lateral wall of the left ventricle (4%), while fibrosis was seen in the septal wall in eight pts (32%). Septal LGE was shown to be a predictor for arrhythmias (Chi^2^ 7.9, *p* = 0.005).

Troponin elevation was also shown to be associated with palpitations. In patients with arrhythmias, mean troponin levels were 10.4 ± 0.8 ng/L, while in patients with no arrhythmias, mean troponin levels were 7.4 ± 0.5 ng/L, and there was a statistically significant difference (*p* < 0.001, Z = −3.3). T1 mapping was not demonstrated to be a predictor for palpitations since there were no statistically significant differences seen between groups with and without palpitations (*p* > 0.05, *p* = 0.122, t = −1.4; mean values of 1100.8 ± 21.4 ms and 1071 ± 9.6 ms, respectively).

In the multivariate analysis, which included troponin, LGE, and native T1 maps, LGE was shown to be the only independent predictor for arrhythmias and palpitation (Chi^2^ 17.2, *p* < 0.001).

## 4. Discussion

CMR is a non-invasive gold-standard method for evaluating cardiac function, structure, and tissue characterization. COVID-19 has now been determined to be a multisystem disease, affecting many parts of the human body. Fatigue and dyspnea have been described to be some of the most common post-COVID-19 symptoms [18,19]. The long-term risks of CCS are still unknown, and the pathophysiology and outcomes of CCS are poorly understood. We hypothesized that CCS may be caused by ongoing myocardial injury and inflammation and proceeded to investigate by CMR.

In our population of symptomatic patients following COVID-19 infection, the percentage of positive images for cardiac damage was lower than what was previously reported by Puntmann et al. [20] (78%) and Huang et al. [21] (58%), but rather similar to Wang et al. [22] (30%). We found evidence to support our hypothesis that CCS in young, previously healthy patients who have had COVID-19 is caused by structural myocardial damage.

In the study by Puntmann [20], 33% of patients had a severe course of disease requiring hospitalization. In this study, most participants with CCS had a mild initial course of COVID-19, but nonetheless, myocardial damage was present in almost 39% of patients. Latent myocardial injury and inflammation could be one of the explanations and possible reasons for prolonged fatigue in these previously healthy patients. It should be noted that in the study by Kravchenko [19], individuals with CCS did not show signs of active myocardial injury or inflammation as determined by cardiac MRI. In the Kravchenko study [19], no connection was found between CCS symptoms and myocardial inflammation induced by COVID-19, despite other reported studies.

This study determined that chest pain is an independent predictor for LGE and myocardial damage. In the literature, there are still no data regarding chest pain and segmental myocardial fibrosis in the LV in patients with myocardial infarction [23] as well as myocardial inflammation. This study also did not find a connection between chest pain and the localization of fibrosis.

We demonstrated in our research that besides clinical symptoms, there are laboratory parameters that could be predictors for myocardial fibrosis. It is well known that cardiac troponins are sensitive and specific markers of myocardial injury. In a study of hospitalized patients with COVID-19, cardiac injury was defined by elevated troponin levels (men > 26 ng/mL, women > 11 ng/mL) and was found in 30% of patients [24]. Elevated troponin levels in this population were associated with a higher in-hospital mortality rate when compared to patients who had normal troponin levels (40.9% vs. 11.1%) [24].

Troponin levels in our population were noted to be an independent predictor for LGE. Troponin was a laboratory parameter that was statistically significantly higher in patients with myocardial LGE (13.1 ± 0.4 ng/L) compared to those with normal myocardium (4.9 ± 0.3 ng/L) with *p* < 0.001, Z = −9.5) and was an independent predictor for LGE (Chi^2^ 112.6, *p* < 0.001, B = −10.6). This study indicates that higher values of troponin are associated with myocardial damage that is seen on CMR as non-ischemic fibrosis in the form of LGE. In conclusion, we could say that in patients with higher values of troponin, we can expect myocardial damage in the form of fibrosis.Also, increased values of troponin were shown to be associated with arrhythmias. Multiple previous studies reported and identified the prognostic significance of troponin elevations in cardiac patients, with higher levels being associated with an increased risk of acute coronary syndromes and heart failure [25,26,27]. Besides predicting poor outcomes in those conditions, this myocardial biomarker has an important prognostic significance in myocarditis.

Patients in this study with cardiac injury presented with more severe acute disease symptoms, especially with a lower blood oxygen saturation level and higher inflammatory biomarkers, including troponin and fibrinogen. Elevated troponin (above 10 ng/L in women and 15 ng/L in men) was detected in 40% of the study population, and elevated fibrinogen levels (above 4 g/L) were found in 17% of the study population. Those values were significantly higher in patients with proven myocardial damage. In Wuhan, elevated hs-troponin was present in 7.2–12% of patients, and around 80% of those with myocardial damage needed intensive care [3]. Cardiac injury in patients has been associated with a lower short-term survival rate compared to patients without myocardial damage, increasing the risk of short-term mortality by nearly two-fold [28].

As for the localization of LGE after COVID-19 infection, most of our patients had infero–postero–lateral subepicardial damage, about 15%, which is quite often and typical in this type of myocarditis (Figure 2). We showed that when compared to other localization of LGE, most patients with septal LGE developed arrhythmias after COVID-19 infection. The types of arrhythmias detected in our patients with septal LGE were supraventricular and ventricular extrasystoles. Some of them had paroxysms of supraventricular tachycardia and ventricle triplets on Holter ECG (Table 3). Also, we should mention that septal fibrosis was seen in eight patients; out of them, six had linear meso-myocardial, continuous LGE, while in two patients, LGE had patchy distribution. Similar findings were presented in an Italian study, ITAMY, in which this septal localization of LGE had the worst outcome in the follow-up of patients [15,29] (Figure 3).

It is known that CMR has prognostic value based on LGE presence, localization, and pattern [15,29,30]. Grän et al. investigated retrospectively 744 patients with suspected myocarditis and reported that MACE (major adverse cardiovascular events) occurred in 15% and LGE was present on CMR in 44% of patients [31]. In terms of the LGE localization and myocardial pattern of LGE, mid-wall and patchy involvement demonstrated a more than two-fold increased hazard to MACE [31]. Septal LGE location showed a strong association with MACE, whereas lateral location did not show a significant association with MACE [31] (Figure 4). In another study that investigated the difference between immune checkpoint inhibitor-induced myocarditis (ICI-M) and virally induced myocarditis, septal LGE was found to be a predictor of MACE in patients with drug-induced myocarditis [32] (Figure 5).

The LV ejection fraction obtained by CMR has been strongly correlated with clinical outcomes after myocardial infarction [25,26]. In clinical practice, LV ejection fraction is still the most substantial predictor of adverse outcomes and represents the basis for further treatment decisions. A recent study of patients with LVEF < 40% and LGE presence found a significantly higher risk for a cardiovascular event in this group compared to those with better ejection fraction [33]. Therefore, it is useful to evaluate both CMR parameters, LVEF and LGE, in assessing cardiovascular risk. However, Sanguineti et al. studied 203 patients with myocarditis based on CMR criteria and found that LVEF was a predictor for MACE in the adjusted analysis, whereas LGE-based variables were not [34]. In our population, ejection fraction was preserved and has not been a predictor for described complaints of patients.

Our patients were treated by the 2021 Guidelines for Management of Hospitalized and Non-Hospitalized Adults With COVID-19 infection. Most of them (75%) required minimal conventional oxygen supplementation and systemic corticosteroid therapy. Additionally, 72.4% of the hospitalized study participants received systemic corticosteroids, around 27.6% (16 of 58 pts) received remdesivir, 5.2% (3 of 58 pts)received convalescent plasma, and 3.4% (2 of 58 pts)received tocilizumab. Pneumonia was noted in 46.5% of the study participants who were treated with antiviral drugs and antibiotics, which comprised almost half of the study population. None of the study participants during hospitalization required inotropic or vasopressor support. Also, some of the patients were treated for hypertension (17.3%) and arrhythmias (13.7%) with beta-blockers and ACE inhibitors. Regarding the vaccination status of our hospitalized patients, only 10% were completely vaccinated with both doses and about 25% with the first dose. Vaccination became more extensive and available in our country after March 2021.

COVID-19, as an infectious disease, has emerged as one of the leading causes of death worldwide, making it one of the most severe public health issues in recent decades. The literature reveals several medications, including remdesivir, hydroxychloroquine, chloroquine, lopinavir, favipiravir, ribavirin, ritonavir, interferons, azithromycin, capivasertib, and bevacizumab, that are used for treatment of COVID-19 infection [35,36]. Older age, male sex, and co-morbidities increase the risk for severe disease. For hospitalized people, 15–30% are going to develop COVID-19-associated acute respiratory distress syndrome (CARDS) [37]. Dexamethasone treatment improves mortality in severe and critical COVID-19 infection, while remdesivir may have modest benefit in time to recovery in patients with severe disease [37,38].

CMR is a potentially helpful diagnostic tool in patients with COVID-19 presenting with myocardial injury and evidence of cardiac dysfunction [33,39]. Late gadolinium enhancement on CMR has been Where no new data were created, or where data is unavailable due to privacy or ethical restrictions, a statement is still required. Suggested Data Availability Statements are available in section shown to be a promising method for improved long-term risk stratification and to evaluate further outcomes [40]. In this context, CMR should be considered to be an important step forward for personalized medicine and treatment. CMR has become a routine clinical and diagnostic method; however, the current prognostic evidence of this approach is still limited.

This study is the first and largest one in Serbia dealing with prognostic clinical and laboratory parameters in COVID-19 patients and the degree of myocardial damage.

## 5. Conclusions

Early detection of acute or chronic sequelae of infection is of utmost importance for determining the optimal therapeutic approach in patients with myocarditis. The use of CMR is a potential risk stratification tool in evaluating outcomes following COVID-19 myocarditis.

### Limitations of This Study

It is important to note several limitations of this study. In each of the patients that were evaluated, this was the first CMR, so we cannot be sure that the reported CMR findings were not also present before SARS-CoV-2 infection. To minimize the role of pre-existing conditions, we included only patients without a history of treatment for cardiac and pulmonary disease. Also, no histopathologic analysis was performed regarding the presence of active myocarditis. However, the quantitative multiparametric MRI techniques that were applied have been reported to provide highly sensitive detection of subclinical myocardial edema and inflammation in post-COVID-19 patients.

## Figures and Tables

**Figure 1 diagnostics-14-00790-f001:**
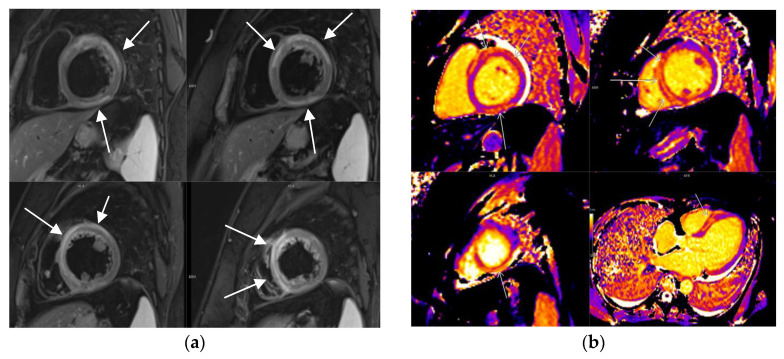
CMR of acute myocarditis demonstrating edema on (**a**) T2w fat-suppression sequence(T2wFS), (**b**) T1 mapping, (**c**) T2 mapping, and (**d**) phase-sensitive inversion recovery sequence (PSIR)—diffuse non-ischemic mesomyocardial LGE.

**Figure 2 diagnostics-14-00790-f002:**
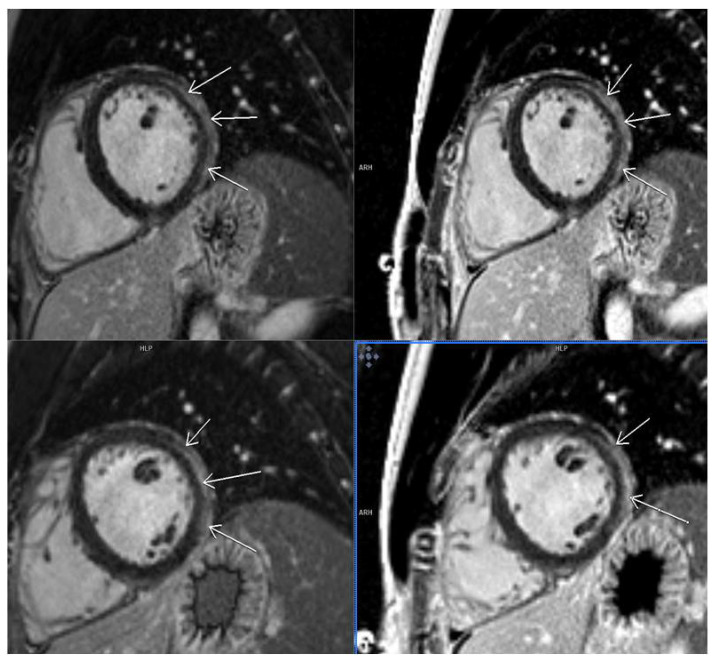
CMR short-axis view (SAX)—LGE in lateral wall basal and mid-ventricle SAX with affection of pericardium.

**Figure 3 diagnostics-14-00790-f003:**
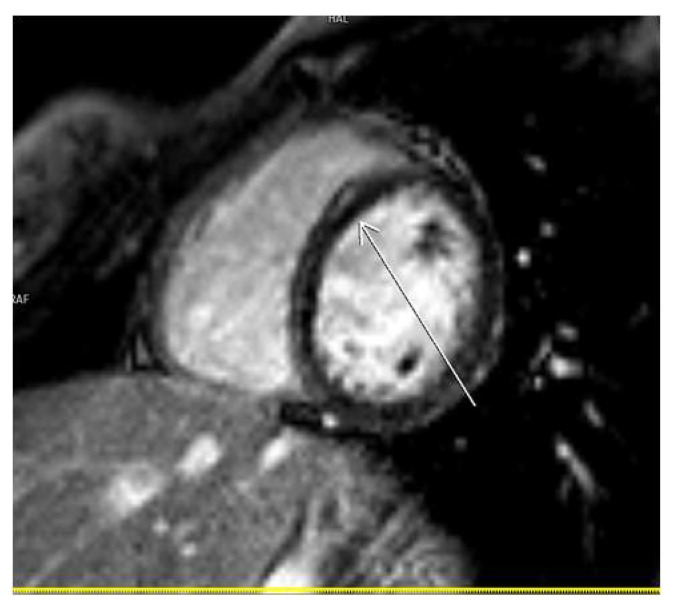
CMR SAX—septal LGE on PSIR sequence.

**Figure 4 diagnostics-14-00790-f004:**
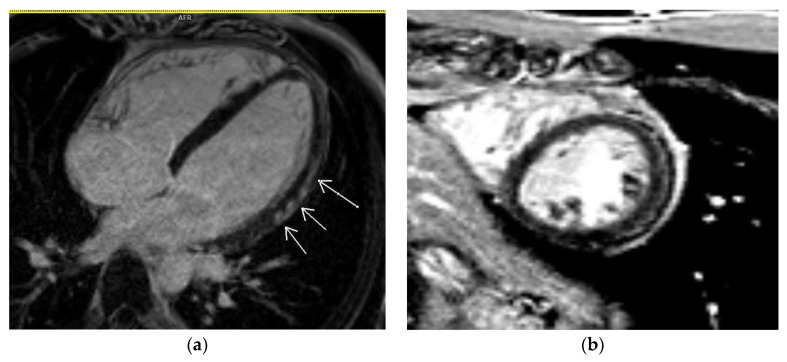
Phase-sensitive inversion recovery sequence (PSIR) demonstrating subepicardial late gadolinium enhancement in the lateral wall of the left ventricle: 4-chamber view (**a**) and mid-SAX (**b**).

**Figure 5 diagnostics-14-00790-f005:**
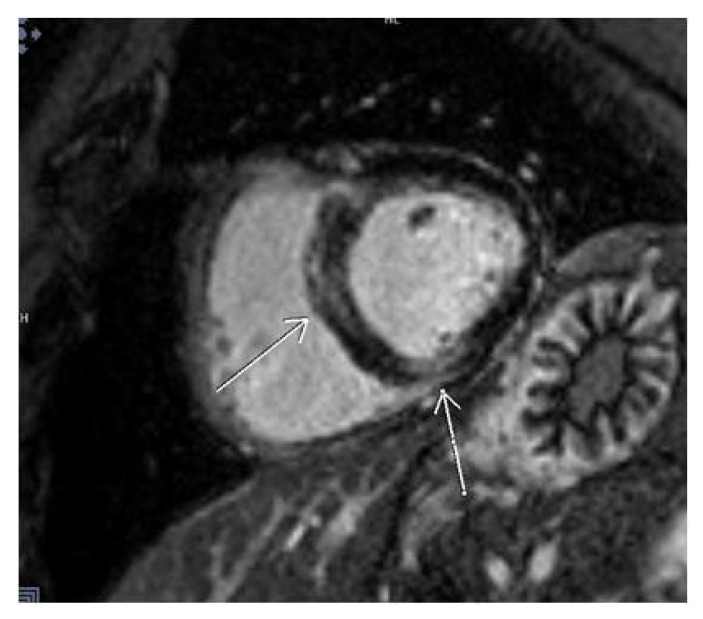
Phase-sensitive inversion recovery sequence (PSIR) demonstrating mesomyocardial late gadolinium enhancement in the septum and inferior wall of the left ventricle (insertion of the septum and right ventricle).

**Table 1 diagnostics-14-00790-t001:** Demographic characteristics and clinical parameters of the study population.

Demographic Characteristics	Result
Age	46.76 ± 15.251
Sex (male/female)	68/71 (48.9%/51.1%)
Days from virus detection to CMR	134.67 ± 80.740
Symptoms of COVID-19	
Chest pain	22 (15.8%)
Fatigue	66 (47.5%)
Arrhythmias	19 (13.7%)
Dyspnea	40 (28.8%)
Loss of taste and smell	24 (17.3%)
Digestive symptoms	31 (22.3%)
Headache	36 (25.9%)
Pneumonia	66 (46.5%)
Co-morbidities	
Dyslipidemia	6 (4.3%)
Hypertension	24 (17.3%)
Diabetes mellitus type II	10 (7.2%)
Smoking	10 (7.2%)
Former smokers	6 (4.3%)
Symptoms after COVID-19	
Chest pain	48 (34.5%)
Fatigue	82 (58.9%)
Arrhythmias	54 (38.8%)
Dyspnoea	58 (41.7%)
ECG changes after COVID-19	
Ischaemic changes	18 (12.9%)
Arrhythmias	14 (10.1%)
Right heart overload	14 (10.1%)
CMR parameters	
LVEF	62.37% ± 4.88%
Patients with LGE	54 (38.8%)
Patients without LGE	85 (61.2%)
Localization of LGE	
Lateral	11 (7.9%)
Septal	9 (6.5%)
Multiple segments (more than 3)	13 (9.4%)
Other localizations	21 (15.1%)

Abbreviations: CMR, cardiac magnetic resonance; LVEF, left ventricular ejection fraction; LGE, late gadolinium enhancement.

**Table 2 diagnostics-14-00790-t002:** Predictors for late gadolinium enhancement.

Laboratory Parameters	LGE-Negative Patients (Mean; SD)	LGE-Positive Patients (Mean; SD)	*p*-Value	Test Value	Independent Predictors for LGE
Troponin values	4.9 ± 0.3 ng/L	13.1 ± 0.4 ng/L	*p* < 0.001	Z = −9.5	Yes
D-dimer levels	0.8 ± 0.1 mg/L FEU	1.8 ± 0.3 mg/L FEU	*p* = 0.005	Z = −3.9	No
Fibrinogen levels	3.2 ± 0.1 g/L	4.3 ± 0.23 g/L	*p* < 0.001	t = −4	No
CK values	143.9 ± 12.4 u/L	112.8 ± 6.9 u/L	*p* = 0.042	Z = −1.6	No
Clinical symptoms					
Chest pain	43.7%	56.3%	*p* = 0.002	Chi^2^ 9.3	Yes
Fatique	39%	61%	*p* < 0.001	Chi^2^ 41.2	Yes
Dyspnea	67.2%	32.8%	*p* = 0.213	Chi^2^ 1.5	No
CMR findings					
T1 mappingvalues	1063.5 ± 9.7 ms	1103.9 ± 17.2 ms	*p* = 0.033	t = −2.2	Yes

Abbreviations: SD, standard deviation; LGE, late gadolinium enhancement; CMR, cardiac magnetic resonance; CK, creatine kinase.

**Table 3 diagnostics-14-00790-t003:** Predictors for arrhythmias.

Laboratory Parameters	Patients with Arrhythmias	Patients without Arrhythmias	*p*-Value	Test Value	Arrhythmia Predictor
Troponin values	10.4 ± 0.8 ng/L	7.4 ± 0.5 ng/L	*p* < 0.001	Z = −3.3	**Yes**
**CMR findings**					
Septal fibrosis	32%	3.4%	*p* = 0.005	Chi^2^ 7.9	**Yes**
T1 mapping	1100.8 ± 21.4 ms	1071.6 ± 9.5 ms	*p* = 0.362	Z = −0.912	No

Abbreviations: CMR, cardiac magnetic resonance.

## Data Availability

The original contributions presented in the study are included in the article material, further inquiries can be directed to the corresponding author.

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
