# Peer review of "Cardiac Magnetic Resonance Imaging as a Risk Stratification Tool in COVID-19 Myocarditis"

_diagnostics, 2024, doi:10.3390/diagnostics14080790_

Round 1

Reviewer 1 Report

Comments and Suggestions for Authors

Nedeljkovic-Arsenovic et al look into late gadolinium enhancement on CMR images and try to find an association with post COVID myocarditis.

Some major and minor concerns are listed below.

“Myocardial injury in hospitalized patients with COVID -19 is accompanied by a worse prognosis”- please ellaborate.

Lines 65,66: remove “year”

Please replace “lab. analysis” with “laboratory analysis”

Line 175: “LGE was found as a sign of myocarditis sequelae”- how long does LGE persist after myocarditis in general? When did your patients have COVID myocarditis? Did they have chronic or subacute myocarditis?

LGE localization chapter: in order to say “predictor”, you need to develop a mathematical model. If you only noticed associations between palpitations and troponin- you need to rephrase accordingly. Developing a predictive mathematical model based on the data ou have available would bring a great plus to this paper.

Lines 251-257: troponin usually signifies myocardial distruction, inflammation or infarction, not fibrosis. Fibrosis does not have a laboratory indicator, it can only be visualised by CMR or biopsy.

“most patients with septal LGE developed arrhythmias after COVID -19 infection”- what types of arrythmia were documented?

Was the septal fibrosis you identified patchy or continuous? Please elaborate.

Please also include data about COVID treatment your patients received. Did they require cardiac support? Was there reason to suspect myocarditis during the actual COVID infection? What was their vaccination status?

I find that in its current form, your paper brings little scientific novelty. I believe it should be enhanced by deeper statistical analysis and describing the types of arrithmias and their association with the site of fybrosis you noticed on CMR.

The article only has 33 references.

Comments on the Quality of English Language

English needs moderate editing.

Author Response

Myocardial injury in hospitalized patients with COVID -19 is accompanied by a worse prognosis [9].

Response: According to the literature data many of patients with COVID-19 infection had myocardial injury. A retrospective single centre study carried out in Wuhan demonstrated significant correlation between myocardial injury and mortality and worse outcome even when compared with patients with previous cardiovascular disease (10). Myocardial involvement in COVID-19 disease is associated with a worse prognosis, but isolated myocarditis is not necessarily a marker of poor prognosis.

Ref 10: Zhou F, Yu T, Du R, Fan G, Liu Y, Liu Z, Xiang J, Wang Y, Song B, Gu X, Guan L, Wei Y, Li H, Wu X, Xu J, Tu S, Zhang Y, Chen H, Cao B (2020) Clinical course and risk factors for mortality of adult in patients with COVID-19 in Wuhan, China: a retrospective cohort

study. Lancet (London, England) 395:1054–1062.

  1. Year was removed from the main text
  2. Replaced lab. with laboratory analysis
  3. Line 175 -LGE was found as a sign of myocarditis sequelae”- how long does LGE persist after myocarditis in general? When did your patients have COVID myocarditis? Did they have chronic or subacute myocarditis?

Response: LGE can persist forever as a sign of a definite fibrosis. It depends of accompanying edema in myocardium. If there is edema in myocardium the bigger chance is that fibrosis disappear. As we already mentioned in a paper our CMR was performed the earliest 15 days after infection and the mean time to do CMR was 135 days, so we do that in majority of patients in subacute and chronic phase.

  1. LGE localization chapter: in order to say “predictor”, you need to develop a mathematical model. If you only noticed associations between palpitations and troponin- you need to rephrase accordingly. Developing a predictive mathematical model based on the data you have available would bring a great plus to this paper.

Troponin elevation was also shown to be associated with palpitations.

  1. Lines 251-257: troponin usually signifies myocardial distruction, inflammation or infarction, not fibrosis. Fibrosis does not have a laboratory indicator, it can only be visualised by CMR or biopsy.

Response: We agree upon this. However, our study indicates that higher values of troponin are associated with myocardial damage that is seen on CMR as a non ischemic fibrosis in a form of LGE. Troponin was laboratory parameter that was statistically significantly higher in patients with myocardial LGE (13.1±0.4 ng/L) compared to those with normal myocardium (4.9±0.3 ng/L) with p<0.001, Z=-9.5 and was independent predictor for LGE (Chi2 112.6, p<0.001, B=-10.6). In conclusion, we could say that in patients with higher values of Troponin we can expect myocardial damage, in these patients - fibrosis.

  1. “most patients with septal LGE developed arrhythmias after COVID -19 infection”- what types of arrhythmia were documented?

Response: The type of arrhythmias detected in our patients with septal LGE were supraventricular and ventricular extrasystoles. Some of them had paroxysms of supraventricular tachycardia and ventricle triplets on Holter ECG.

  1. Was the septal fibrosis you identified patchy or continuous? Please elaborate.

Response: Septal fibrosis was seen in 8 pts, out of them 6 had linear meso-myocardial, continuous LGE, while in two patients LGE had patchy distribution.

  1. Please also include data about COVID treatment your patients received. Did they require cardiac support? Was there reason to suspect myocarditis during the actual COVID infection? What was their vaccination status?

Response: Our patients were treated by the 2021. Guidelines for Management of Hospitalized and Non-Hospitalized Adults With COVID-19 infection. Most of them (75%) required minimal conventional oxygen supplementation and systemic corticosteroid therapy. Additionally, 72.4% of the hospitalized study participants received systemic corticosteroids, around 27.6% (16 of 58 pts) have got remdesivir, 5.2% (3 of 58 pts) convalescent plasma, and 3.4% (2 of 58 pts) tocilizumab. Pneumonia was noted in 46.5% treated with those antiviral drugs accompanied by antibiotics that received almost half of study participants. None of the study participant during hospitalization required inotropic or vasopressore support. Also some of the patients were treated for hypertension (17,3%) and arrhythmias (13.7%) with beta blockers and ACE inhibitors. As far as for the vaccinated status of our hospitalized patients only 10% were completed vaccinated with both doses and about 25% with first dose. Vaccination became more extensive and available in our country after March/April 2021. year.

COVID-19 as an infectious disease has been emerged for one of the leading causes of death worldwide, making it one of the severe public healthy issues in recent decades. The literature reveals several medications, including remdesivir, hydroxychloroquine, chloroquine, lopinavir, favipiravir, ribavirin, ritonavir, interferons, azithromycin, capivasertib and bevacizumab, that are used for treatment of COVID-19 infection. [36, 37]. Older age, male sex, and co-morbidities increase the risk for severe disease. For hospitalized people 15-30% are going to develop COVID-19 associated acute respiratory distress syndrome (CARDS) [38]. Dexamethasone treatment improves mortality in severe and critical COVID-19 infection, while remdesivir may have modest benefit in time to recovery in patients with severe disease. [38, 39].

References:

  1. Fatima U, Rizvi SSA, Raina N, Fatima S, Rahman S, Kamal MA, Hassan MI. Therapeutic Management of COVID-19 Patients: Clinical Manifestation and Limitations. Curr Pharm Des. 2021;27(41):4223-4231. doi: 10.2174/1381612826666201125112719.
  2. Wahab S, Ahmad MF, Hussain A, Usmani S, Shoaib A, Ahmad W. Effectiveness of Azithromycin as Add-on Therapy in COVID-19 Management. Mini Rev Med Chem. 2021;21(19):2860-2873. doi: 10.2174/1389557521666210401093948.
  3. Attaway AH, Scheraga RG, Bhimraj A, Biehl M, Hatipoğlu U. Severe covid-19 pneumonia: pathogenesis and clinical management. BMJ. 2021 Mar 10;372:n436. doi: 10.1136/bmj.n436.
  4. Perveen RA, Nasir M, Talha KA, Selina F, Islam MA. Systematic review on current antiviral therapy in COVID-19 pandemic. Med J Malaysia. 2020;75(6):710-716.

Reviewer 2 Report

Comments and Suggestions for Authors

This paper focuses on validating the effectiveness of MRI as Risk Stratification Tool in COVID19 Myocarditis, it is interesting and can be publishable. My concerns are as follows,

1.      For the task of image analysis, the Argus is employed. This software is developed about 20 years ago, see the following ref.,

T. Odonnell, G. Funka-lea, H. Tek, M.-P. Jolly, and M. Rasch, Comprehensive Cardiovascular Image Analysis Using MR and CT at Siemens Corporate Research, International Journal of Computer Vision 70(2), 165178, 2006

How about the segmentation results? And how does it affect the final results?

Today, the deep learning methods have achieved promising results for image segmentation, such as the following works, it is better if the author can say something about this topic.

C. Zotti, Z. Luo, A. Lalande, and P.-M. Jodoin, Convolutional Neural Network With Shape Prior Applied to Cardiac MRI Segmentation, IEEE Journal of Biomedical and Health Informatics, vol. 23, no. 3, pp. 1119–1128, 2019

X. Sun, L.-H. Cheng, S. Plein, P. Garg, and R. J. van der Geest, Transformer Based Feature Fusion for Left Ventricle Segmentation in 4D Flow MRI, MICCAI 2022

Context-aware network fusing transformer and V-Net for semi-supervised segmentation of 3D left atrium,” Expert Systems with Applications, vol. 214, p. 119105, 2023

2.      It seems that the sample size and distribution are not enough, only 139, and “without known cardiac or pulmonary diseases”, it is better if the authors could say something more about this topic.

 In addition, what is “lab. Analyses”  in the abstract? 

Author Response

. For the task of image analysis, the Argus is employed. This software is developed about 20 years ago, see the following ref.

Response: Yes that software has been used for our image analysis. We will take this reference into account. Thank you for advice.

Ref 14: T. Odonnell, G. Funka-lea, H. Tek, M.-P. Jolly, and M. Rasch, Comprehensive Cardiovascular Image Analysis Using MR and CT at Siemens Corporate Research, International Journal of Computer Vision 70(2), 165–178, 2006.

  1. How about the segmentation results? And how does it affect the final results?

Response: In our population, 85 pts (61.2%) had a normal myocardium on CMR, while in 54 patients (38.8%)  had LGE. We didn’t get statistical significance in any separate segment of left ventricle since the number of patients in those subgroups was insufficient.  Therefore, in order to increase the sample size and according to the clinical practice, we divided patients into groups by localization of LGE distribution in the left ventricle wall (septal wall, lateral wall, more than 3 segments, and other localization).  Segmentation was done manually in this study, although in the study of Zotti C. et al. (16)  promising results were reported using neural networks, both regarding the accuracy and time of processing. Also automatic segmentation of the left atrium (LA) in magnetic resonance images is of great significance for studying structure of left atrium in order to facilitate the diagnosis and treatment of atrial fibrillation (17).

Ref 16. C. Zotti, Z. Luo, A. Lalande, and P.-M. Jodoin, Convolutional Neural Network With Shape Prior Applied to Cardiac MRI Segmentation, IEEE Journal of Biomedical and Health Informatics, vol. 23, no. 3, pp. 1119–1128, 2019.

  1. Chenji Zhao, Shun Xiang, Yuanquan Wang, Zhaoxi Cai, Jun Shen, Shoujun Zhou, Di Zhao, Weihua Su, Shijie Guo, Shuo Li. Context-aware network fusing transformer and V-Net for semi-supervised segmentation of 3D left atrium,” Expert Systems with Applications 2022, 214(6):119105.

  1. It seems that the sample size and distribution are not enough, only 139, and “without known cardiac or pulmonary diseases”, it is better if the authors could say something more about this topic.

Response: Our study included patients who were previously healthy individuals without any treated and known cardiac or pulmonary disease. They had COVID-19 infection and persistent symptoms following COVID-19 infection. Those symptoms were predominantly  cardiovascular symptoms such as of persistent fatigue, exertional dyspnea, chest pain and arrhythmias. Persistent clinical symptoms were the main reason for CMR in order to evaluate the presence of myocardial damage or myocarditis as a possible explanation of mentioned symptoms. In our country this was the first study and the biggest one to deal with CMR and COVID 19 patients.

Round 2

Reviewer 1 Report

Comments and Suggestions for Authors

I feel this paper quality has improved. However, one concern still remains- please provide patient consent for publication.

I understand patients have signed an informed consent to undergo the CMR, but this does not imply they agreed to publication.

Comments on the Quality of English Language

Minor English editing is still required.

Author Response

In the attachment you will find singed patient consent for publication as a PDF file. Even though we have already sent that to Assistant editor Ms Carol at the very beginning.

Thank you very much for the comments that have significantly contributed to the overall quality of the paper.

If you need anything else we are at you disposal.  
